# Effects of Composite Attachments on Orthodontic Clear Aligners Therapy: A Systematic Review

**DOI:** 10.3390/ma15020533

**Published:** 2022-01-11

**Authors:** Riccardo Nucera, Carolina Dolci, Angela Mirea Bellocchio, Stefania Costa, Serena Barbera, Lorenzo Rustico, Marco Farronato, Angela Militi, Marco Portelli

**Affiliations:** 1Department of Biomedical and Dental Sciences and Morphofunctional Imaging, Section of Orthodontics, University of Messina, 98125 Messina, Italy; riccardo.nucera@unime.it (R.N.); angelamirea@live.it (A.M.B.); stefaniacosta94@gmail.com (S.C.); serena.barbera93@gmail.com (S.B.); lorenzo.rustico@gmail.com (L.R.); amiliti@unime.it (A.M.); 2Department of Biomedical, Surgical and Dental Sciences, Fondazione IRCCS Cà Granda—Ospedale Maggiore Policlinico, University of Milan, 20122 Milan, Italy; carolinadolci3@gmail.com (C.D.); marcofarronato@msn.com (M.F.)

**Keywords:** invisible orthodontics, clear aligner therapy, clear aligners, invisalign, attachments, auxiliary elements, systematic review

## Abstract

This systematic review aims to highlight the differences between different clear aligner therapies that differ in the presence of attachments or in attachment configuration. Eight electronic databases were searched up to March 2020. Two authors independently proceeded to study selection, data extraction, and risk of bias assessment. The analysis of the results was carried out examining six groups of movements (mesio-distal tipping/bodily movement; anterior bucco-lingual tipping/root torque; posterior bucco-lingual tipping/expansion; intrusion; extrusion; rotation). Five clinical trials were selected and all of them showed a medium risk of bias. Literature showed that attachments mostly increase the effectiveness of orthodontic treatment with clear aligners, improving anterior root torque, rotation, and mesio-distal (M-D) movement; they are also important to increase posterior anchorage. However, some articles showed contradictory or not statistically significant results. Attachments also seem to improve intrusion, but the evidence about this movement, as well as extrusion, is lacking. No studies evaluated posterior bucco-lingual tipping/expansion. Further clinical trials are strongly suggested to clarify the influence of attachments and their number, size, shape, and position on each orthodontic movement.

## 1. Introduction

In the past decades, the demand of an aesthetic alternative to conventional fixed devices, especially by adult patients, has oriented the research toward the development of more comfortable and aesthetic appliances, leading to the development of clear aligner therapy [1,2,3,4]. Thermoplastic appliances have thus become popular worldwide and many researchers have focused their interest in this field [5,6,7,8,9,10,11]. As a result of new materials and technologies, aligners have been continually improved in many aspects and they are currently used in an increasing number of cases [12,13]. As previously reported in the literature, clear aligner therapy often requires the use of auxiliaries (attachments, altered aligner geometries, inter-arch elastics, etc.,) to improve the efficacy of orthodontic movement [13,14,15]. Attachments are force transducers that seem to improve the biomechanics of invisible aligners. Essentially, attachments are a protrusion of composite material polymerized onto tooth surface, applied in order to improve aligner retention and to obtain orthodontic movements previously considered critical to achieve. They are able to reach these goals through an enhancement of the mismatch in specific points, an improvement of the contact area, and a better force system application. Attachments can have different shapes, designed for specific tasks and/or specific dental movements. Literature showed that the combination between disposition, shape, size, and number of attachments can greatly influence the efficacy of orthodontic treatment [4,16]. In this context, a better understanding of forces and moments generated by different attachments and the knowledge of biomechanics principles are essential in order to select proper attachments and, ultimately, to improve efficacy and efficiency of orthodontic treatment. The aim of this systematic review is to highlight the differences between orthodontic cases treated with and without attachments and to clarify what is the best shape, size, number, and position of attachments for each specific orthodontic movement.

## 2. Materials and Methods

### 2.1. Protocol

This systematic review was conducted following the guidelines of the Cochrane Handbook for Systematic Reviews of Interventions (version 5.1.0) [17] and it is reported according to the PRISMA statement [18]. The protocol of this review was preliminarily published at the following web address: https://www.crd.york.ac.uk/prospero/display_record.php?ID=CRD42020150671 on 3 March 2020 with the registration number CRD42020150671.

### 2.2. Research Strategy and Information Sources

A search of articles published up to March 2020 concerning the use of attachments in orthodontic therapy with clear aligners was carried out by means of several electronic databases: PubMed, Embase, Cochrane—Database of Systematic Reviews, Cochrane—Central Register of Controlled Trials, Web of Science, Lilacs, ClinicalTrials.gov (accessed on 3 March 2020) and Proquest. Grey literature was also searched through OATD (Open Access Theses and Dissertations), while no manual search was conducted. All eligible articles for inclusion were manually reviewed. Systematic reviews and meta-analyses on this subject were also identified and their reference lists scanned for additional studies. An adjusted search strategy was performed for each of the eight consulted databases. All electronic searches were conducted between 10 January and 22 March 2020. The search strategies for each individual database are reported in Table 1. No language or publication year restrictions were applied.

### 2.3. Selection of Studies

The inclusion criteria were:Studies involving the presence of composite attachments in orthodontic therapy with clear aligners;Clinical studies on humans with a control group without attachments and/or a comparison between different configurations of composite attachments.


The exclusion criteria were:
Studies that do not relate to the topic or are related but have a different purpose;Clinical studies on humans without a control group and without a comparison between different configurations of attachments;Case reports, experimental studies that do not include humans, posters, books.

Duplicate references were excluded; articles with the same sample and reporting the same outcomes were considered only once. Two review authors (C.D. and L.R.) independently performed the preliminary study selection based on titles and abstracts evaluation. The same two reviewers then proceeded to the analysis of the full text, in order to finally select the included articles. The results of the two independent study selections were compared. Cohen’s Kappa was calculated to determine the concordance between the two reviewers. Any disagreement was resolved after consulting a third author [R.N.], in order to reach a unanimous consensus.

### 2.4. Data Extraction Process

Two authors (C.D. and L.R.) autonomously extracted study characteristics after creating a specific extraction form, which included study design, setting, analyzed sample, auxiliary elements taken into account, methods of analysis, performed movements, treatment duration as well as outcomes. The agreement between the two authors for study characteristics was assessed by calculating the percentage of consistent extraction data and any disagreement was solved by discussion with another author (R.N.).

### 2.5. Assessment of Risk of Bias

Two authors (C.D. and L.R.) independently analyzed every potential source of bias, using the Downs and Black scale. It consisted of 27 questions evaluating: reporting [10 questions], external validity [3 questions], internal validity or bias [7 questions], internal validity or confounding or selection bias [6 questions] and power [1 question]. According to this scale, answers were scored from 0 to 1 point, except for two of them: in fact, in correspondence with question number 5, the value can vary from 0 to 2, while in question number 27 it can vary from 0 to 5 [19]. As far as regarding the latter, however, we simplified the assessment of this question by scoring this answer at 0 or 1 point, giving 1 point for a preliminary power analysis calculation. The total maximum score for a clinical study was therefore equal to 28 [19]. To establish the risk of bias, we therefore created a scale of values, defining a study at high risk if it had a total score between 0 and 8, at medium risk if it was between 9 and 18 and at low risk if it was between 19 and 28. Any disagreement between the two authors was resolved after consulting another author (R.N.). The level of agreement for risk of bias evaluation was assessed with Cohen Kappa statistics matching the results of the two evaluating authors (C.D. and L.R.).

### 2.6. Data Analysis

Given the heterogeneity of the included studies, it was decided to avoid the execution of a meta-analysis: as a consequence, results were mainly evaluated from a qualitative point of view. For this purpose, six groups of movements were considered: mesio-distal tipping/bodily movement, anterior bucco-lingual tipping/root torque, posterior bucco-lingual tipping/expansion, intrusion, extrusion and rotation. Taking into consideration the aim of this review, the analysis of the results was carried out with the intent to highlight the effectiveness of attachments in orthodontic therapy with clear aligners and to clarify how attachments shape, size, number, and position potentially affect clinical outcomes.

## 3. Results

### 3.1. Search Results and Articles Selection

Table 1 shows the performed electronic searches, providing the following information for each search: electronic database, date of search, search strategy, and number of retrieved items. Electronic searches have identified 3959 studies; one study has been retrieved from external sources. 3674 articles remained after the duplicates removal and were examined on the basis of title and abstract. After this first screening, 3648 articles were excluded and the remaining 26 were examined for eligibility based on the full text. At the end of this further selection 5 studies were identified. Figure 1 reports the selection flow chart according to the PRISMA guidelines.

### 3.2. Study Characteristics

The characteristics of the five selected studies are summarized in Table 2. They included two randomized clinical trials (RCTs), two controlled clinical trials (CCTs), and a case series. Moreover, the studies took place in universities, public hospitals, and private practices. The RCTs included a total of 116 patients [16,20], the CCTs considered a total of 57 patients [21,22], the case series study comprised a sample of 30 patients [23]. All the trials included both sexes, with the exception of one trial where the authors did not specify this information [20]. Studies were performed in adult patients, with a mean age ranging between 18.0 y.o. [20] and 32.9 y.o. [21]. Treatment duration was approximately two years in three clinical trials [16,20,23], while one study reported on average 7.2 months of treatment [22] and one article just specified the average number of aligners used for the entire treatment [eighteen] without reporting the wearing period per each aligner [21]. All the included studies comprised the presence of at least one control group without attachments or with a different attachment configuration or with the use of additional auxiliary elements. Most of the studies assessed the effectiveness of aligner therapy by comparing the predicted and the obtained clinical results [by digital models overlapping], with the exception of one study that compared pre-treatment and post-treatment cephalometric values [16].

The analyzed attachments were: rectangular [16,20,22,23], rectangular beveled [21], optimized [21,23], and ellipsoid [20,21,22]. Studies evaluated different dental movements: three articles analyzed mesio-distal tipping or bodily movement [16,21,23], three studies assessed anterior bucco-lingual tipping/root torque [16,21,23], two reports dealt with intrusion and/or extrusion [20,23], and finally three research considered rotation [20,21,22]; no one analyzed posterior bucco-lingual tipping/expansion.

### 3.3. Risk of Bias Assessment

Five clinical studies were considered, according to the modified Downs and Black scale tool, at a medium risk of bias [16,20,21,22,23]. The average score was 16 out of 28, with a minimum and maximum score of 12/28 and 18/28 respectively (Table 3).

The agreement between the two review authors regarding study selection and risk of bias assessment was adequate, with Cohen kappa ranging between values of 0.89 and 0.96.

### 3.4. Analysis of the Results

The analysis of the results was carried out by individually examining six groups of movements [mesio-distal tipping/bodily movement; anterior bucco-lingual tipping/root torque; posterior bucco-lingual tipping/expansion; intrusion; extrusion; rotation], as summarized in Table 4. No article assessed posterior bucco-lingual tipping/expansion. The effects of auxiliary elements different from attachments [such as divots and power arms] were not evaluated.

## 4. Discussion

Based on our current knowledge, this is the first systematic review that evaluates the influence of composite attachments on orthodontic therapy with transparent aligners and the possible differences between their configurations (shape, size, number and/or position). In order to obtain a more schematic analysis of the results, we separately assessed five groups of movements (anterior B-L tipping/root torque, intrusion, extrusion, rotation, and bodily movement in M-D direction). No article analyzed the effects of attachments on posterior B-L inclination and/or expansion movement.

While B-L tipping is considered an easier movement to be obtained [24,25], anterior root torque represents a challenge for treatments with aligners [26,27]. This review highlights the importance of auxiliary elements to achieve a better root control, a concept previously reported [12,28]. Simon demonstrated that torque [as well as bodily movement] can be obtained with aligners and auxiliary elements, such as attachments and power ridges, since they are able to release an adequate system of forces [21]. More specifically, the incisor torque is smaller with the use of horizontal ellipsoidal attachments in comparison with power ridges, which provide a force closer to the tooth neck, are easier to apply and more aesthetic and, finally, increase aligner resistance at the gingival third [21]. However, literature showed that attachments and power ridges may not be sufficient to ensure a right root control and hypercorrection or refinement might be necessary [21], as previously suggested by Kravitz [22] and more recently by Houle [29] and Khosravi [30]. The retraction of anterior teeth with a proper root control also depends on the achievement of a suitable posterior dental anchorage [16,23], which can be improved by adding attachments on a greater number of teeth [from canine to second molar] [13,16,31].

Literature shows that both intrusion and extrusion can be facilitated by the use of attachments [15,32,33,34]. Durrett [20] confirmed these findings analyzing the intrusion of incisors, canines, and premolars. In his study all the groups with attachments showed greater efficiency than the control group without attachments performing intrusion movement. He noticed no significant differences among the analyzed attachment shapes. The authors of this clinical trial highlighted that several possible limitations could have affected their results, so these findings should be confirmed in the future. Moreover, attachments could improve intrusion by enhancing the accuracy of the fit: some authors suggested to use attachments on premolars in order to enhance the retention of aligners during intrusion [33,34]. This effect could be useful in deep bite cases in order to improve levelling of the Spee curve [33]. The intrusion movement was also considered by Dai et al. [23]. However, the aim of the study and the analyzed teeth were different from Durrett’s study [20]; Dai compared predicted and achieved tooth movement of maxillary first molars and central incisors in extraction cases. The results of this trial showed greater intrusion of posterior teeth, compared to the predicted virtual tooth movement. As far as regarding the influence of attachments, the group with the optimized G6 attachments had the greatest difference between predicted and achieved tooth movement. These authors proposed that these treatment outcomes could be related to the occlusal splint effect that is created wearing the aligner and they suggest to consider heavy occlusal contacts on posterior teeth during setup, in order to prevent posterior open-bite [23].

Extrusion seems to be one of the most critical movements to be carried out by means of aligners [especially when referred to central incisors], due to the lack of elastic deformation of the aligner in the vertical direction [8,9,11]. Several authors in literature have highlighted this critical issue [8,9,11]. Unfortunately, Durrett’s study does not allow to obtain clear conclusions about this movement, due to the small sample size [20]. However, a recent experimental study demonstrated, through FEM analysis, that the use of one attachment bonded on the palatal side could improve incisor extrusion [35]: this is an encouraging sign for orthodontic research and future clinical trials could be useful to confirm these results.

Rotation is considered one of the most difficult movement to correct with transparent aligners, in particular when it involves conical teeth. Literature shows that the use of attachments could increase the effectiveness of derotation movement, creating undercuts and improving retention [2,36,37,38,39]. However, two of the five clinical studies included in this review do not highlight significant differences between the treated groups with and without attachments [21,22]. Kravitz attributed the absence of evident advantages to a large number of canines subjected to a rotation greater than 5° within the attachment group. Another factor that could justify the results reported in this article is the small sample size of the attachment group. It is worth considering that different patients’ compliance among the groups could affect the final outcomes [21]. Finally, a third study shows conflicting results [20]. In the latter work, attachments caused an improvement of rotation in the “reboot” group and a worse clinical outcome in the “non-reboot” group, compared to the control one without attachments [20]. However, the small sample size and other study limitations could have affected the study results, as also stated by the author of this clinical trial. The confounding factors of these studies do not allow us to draw clear conclusions about the efficacy of attachments in derotation and further clinical studies are needed to evaluate this. With reference to the number of attachments, Durrett showed that two attachments do not improve the magnitude of rotation, showing worse outcomes than the other attachment groups or resulting even less effective compared to the control group without attachments [20]. This result was confirmed by a subsequent experimental study by Momtaz [4], but further experimental and clinical studies are needed to confirm this evidence. As regards shape and size, larger attachments with sharper edges seem to perform better during derotation movements [20]. It is worth noting that there are other factors that must be taken into account during a treatment plan which includes rotation: the amount of total derotation movement [21,38], staging [degree of derotation per aligner] [21,37], interproximal reduction [IPR] [22], and the use of buttons with elastics [20] can in fact influence derotation efficacy.

Bodily mesio-distal movement is also considered difficult to achieve with aligners [27,40,41]. Nowadays, the introduction of new techniques and auxiliaries allows a better root control [13]; technological innovations allowed an improvement in orthodontic dental movement even with traditional and self-ligating multibrackets appliances [42,43]. Some evidence showed that aligners with attachments are able to release the necessary force system in order to achieve bodily molar distalization [16,21] and that staging plays an important role in achieving treatment success [21,36,44]. Attachments should be able to create a moment useful to counteract dental tipping: this moment seems to be determined by a complex force system on attachments active surfaces [3]. Some FEM analysis, for example, demonstrated that movements like canine distalization or incisor bodily movement during diastema closure can be improved by attachments [3,45,46,47]. However, clinical studies do not allow to draw clear conclusions about the attachment capacity to improve this movement. Simon et al. showed that the attachment group could be more effective than the group without attachments, but the differences do not seem to be clinically significant: the mean accuracy of movement obtained with and without attachments was 88.4% and 86.9%, respectively [21]. As far as concerning the number of attachments, it seems that the use of five attachments per quadrant [from canine to second molar] could improve posterior anchorage, molar bodily movement and it could enhance the amount of intrusion [16]. Posterior anchorage could also be influenced by the shape of attachments: optimized and rectangular horizontal attachments have shown the best results in molar anchorage, unlike the vertical rectangular ones, which were the least effective [23]. However, the sample size analyzed for vertical attachments was small: further studies are needed to compare different attachment shapes [23]. Considering the available evidence, further clinical trials are needed to evaluate the influence of attachments in the efficacy of molar distalization and posterior anchorage.

This systematic review highlights that attachments often cause significant therapeutic effects during aligner treatments. This consideration allows us to conclude that attachments could improve aligner biomechanics and they should consequently be considered important auxiliary elements of aligner therapy. The limitations of this review are strictly related to the limitations of the included studies. Literature on this topic showed big heterogeneity; there are a few clinical studies and only two of them performed randomization during patients’ allocation. These study characteristics were used to score at Level IV the overall level of evidence of this review, according to a validated approach [48,49]. On the other hand, included trials were labelled with a medium risk of bias and none of them showed a high risk. More clinical studies are therefore needed to better define the role of attachments on clear aligner therapy and to clarify which is the best configuration for each movement.

## 5. Conclusions

The anterior root torque can be improved by the use of auxiliary elements, such as power ridges and attachments. However, they may be insufficient to ensure a right root control. Posterior anchorage seems important to ensure a greater control during anterior teeth retraction. It can be improved by adding attachments on a greater number of teeth [from canine to second molar].The evidence of the influence of attachments on intrusion and extrusion is lacking, although attachments seem to improve intrusion. No clinical studies evaluated posterior bucco-lingual tipping/expansion.Conflicting results were found about the ability of attachments to improve tooth rotation control. The majority of the studies showed a positive influence of attachments on derotation movement, although not statistically significant. The use of two attachments on the buccal and palatal side or the addition of attachments on adjacent teeth do not seem to improve rotation. Larger attachments with sharper edges showed better outcomes. However, several factors seemed to influence derotation effectiveness.The results have shown that the use of attachments could increase the molar mesio-distal movement efficacy. However, this improvement may not be clinically significant. Posterior anchorage can be improved by increasing the number of attachments bonded on the posterior teeth and optimized and rectangular horizontal attachments have shown the best results.Further clinical studies will be necessary to confirm all the above reported findings and to increase knowledge about the influence of attachments on different types of movement.

## Figures and Tables

**Figure 1 materials-15-00533-f001:**
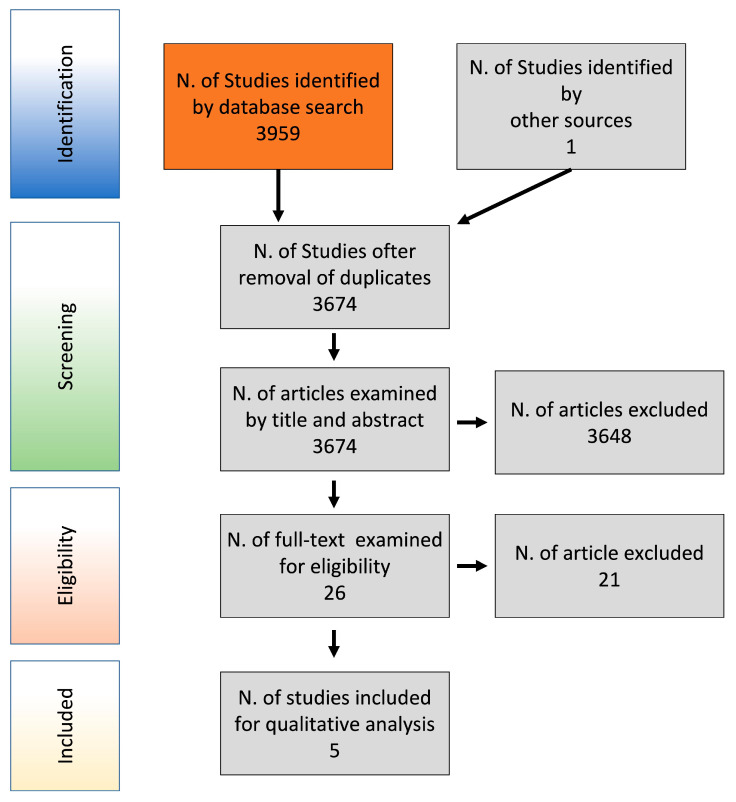
Flow chart of studies selection.

**Table 1 materials-15-00533-t001:** Consulted databases, applied search strategy, and numbers of retrieved studies.

Database of Published Trials	Search Strategy Used	Hits
MEDLINE searched via Pubmed Searched on 22 March 2020 via https://www.ncbi.nlm.nih.gov/pubmed/	(((((((((orthodontic*) OR treatment*) OR therapy) OR therapies) OR appliance*) OR device*)) AND (((((attachment*) OR accessory) OR accessories) OR auxiliary) OR auxiliaries)) AND ((((((((aligner*) OR Invisalign) OR thermoplastic) OR transparent) OR clear) OR invisible) OR thermoform*) OR removable))	1822
COCHRANE Database of Systematic Reviews searched via The Cochrane Library Searched on 22 March 2020 via https://www.cochranelibrary.com/	((orthodontic*) OR (treatment*) OR (therapy) OR (therapies) OR (appliance*) OR (device*)) AND ((aligner*) OR (Invisalign) OR (thermoplastic) OR (thermoform*) OR (transparent) OR (invisible)) AND ((attachment*) OR (accessory) OR (accessories) OR (auxiliary) OR (auxiliaries))	666
COCHRANE Central Register of Controlled Trials searched via The Cochrane Library Searched on 22 March 2020 via https://www.cochranelibrary.com/	((orthodontic*) OR (treatment*) OR (therapy) OR (therapies) OR (appliance*) OR (device*)) AND ((aligner*) OR (Invisalign) OR (thermoplastic) OR (thermoform*) OR (transparent) OR (invisible)) AND ((attachment*) OR (accessory) OR (accessories) OR (auxiliary) OR (auxiliaries))	34
WEB OF SCIENCE Searched on 22 March 2020 via https://www.webofknowledge.com/	(((((((((orthodontic*) OR treatment*) OR therapy) OR therapies) OR appliance*) OR device*)) AND ((((((aligner*) OR Invisalign) OR thermoplastic) OR thermoform*) OR transparent) OR invisible)) AND (((((attachment*) OR accessory) OR accessories) OR auxiliary) OR auxiliaries))	344
LILACS Searched on 22 March 2020 via http://lilacs.bvsalud.org/	(tw:(aligner OR Invisalign OR thermoplastic OR thermoformed OR transparent OR invisible)) AND (tw:(attachment OR accessory OR accessories OR auxiliary OR auxiliaries))	7
EMBASE Searched on 10 January 2020 via https://www.embase.com/	(orthodontic* OR treatment* OR ‘therapy’/exp OR therapy OR therapies OR appliance* OR device*) AND (aligner* OR Invisalign OR ‘thermoplastic’/exp OR thermoplastic OR thermoform* OR transparent OR invisible) AND (attachment* OR accessory OR accessories OR auxiliary OR auxiliaries)	269
CLINICALTRIALS.GOV Searched on 22 March 2020 via https://clinicaltrials.gov/	(aligner OR Invisalign OR thermoplastic OR thermoformed OR transparent OR invisible) AND (attachment OR accessory OR accessories OR auxiliary OR auxiliaries)	43
PROQUEST Searched on 10 January 2020 via https://www.proquest.com/	(aligner OR Invisalign OR thermoplastic OR thermoformed OR transparent OR invisible) AND (attachment OR accessory OR accessories OR auxiliary OR auxiliaries) AND Orthodontic*	774
TOTAL		3959

**Table 2 materials-15-00533-t002:** Characteristics of included clinical trials.

Study	Type of Study	Setting	Analyzed Sample	Auxiliary Elements	Analysis Methods	Performed Movements	Duration of Treatment	Outcome
Dai et al. [23]	Case Series	The Second Dental Center, Peking University School and Hospital of Stomatology	30 patients (4 M–26 F) Age: 19.4 ± 6.3 First premolar extraction cases treated with Invisalign. Three variables considered: -Age -Type of attachment -Initial crowding	On first molar: -Vertical attachment (3mm) -Horizontal attachment (3mm) -Horizontal attachment (5mm) -G6 optimized attachment	Superimposition of: -Real and virtual pre-treatment models -Real pre- and post-treatment models -Virtual pre- and post-treatment models -All four models	On first molar: -M-D TIPPING -M-D TRANSLATION -O-G TRANSLATION On central incisor: -TORQUE -V-L TRANSLATION -O-G TRANSLATION	22.3 ± 4.6 months Aligner change every 1–2 weeks	- Difference between predicted and achieved tooth movement in maxillary first molar and central incisor -Influence of age, initial crowding and type of attachment
Durrett [20]	RCT	University of Florida Orthodontic Research clinic	99 patients reduced to 86 Age: 18+ 6 configurations: -5 groups with different attachments -1 control group (without attachments) 2 groups: -Extraction and non-extraction cases	-No attachments (Group C) -Attachments with different shapes on the buccal surface (Groups A-B-D-F) -Attachments bonded to both buccal and lingual surfaces (Group E)	Superimposition of: initial models and final models or first reboot	-ROTATION of canine and premolar -INTRUSION and EXTRUSION of incisor, canine and premolar	Minimum 2 years Aligner change every 2 weeks	-ROTATION: Comparison between predicted and achieved tooth movement in reboot and non-reboot cases -INTRUSION and EXTRUSION: Comparison between predicted and achieved tooth movement in reboot cases
Garino et al. [16]	RCT	Orthodontic clinics in Turin and Vancouver Collections of the AAO Foundation Craniofacial Growth Legacy	30 patients with class II malocclusion (12 M–18 F) Age: 30.5 3 configurations: -Group C1—5 attachments per quadrant -Group C2—3 attachments per quadrant -Group C—Control	Vertical rectangular attachments -Group C1—From maxillary canine to second molar -Group C2—From maxillary first premolar to first molar Class II elastics in the first phase Power ridge in the second phase	Superimposition of T0 and T1 cephalograms	-Maxillary molars DISTALIZATION -Incisors RETRACTION	Average time:24.3 months Aligner change every 2 weeks	Comparison of the position of upper molars and central incisors between T0 and T1 (angular, horizontal and vertical measurements expressed as angles and distances from y-axis, x-axis and occlusal plane).
Kravitz et al. [22]	Prospective CCT	Department of Orthodontics—University of Illinois-Chicago	38 pazients reduced to 31 (13 M–18 F) Average age: 29.4 3 configurations: -Attachment only group (AO) -Interproximal-reduction only group (IO) -Group without attachments (N)	Attachemnt Only group: -Vertical or horizontal ellipsoid attachments -Horizontal rectangular attachments	Superimposition of the final stage of the pre-treatment model (ClinCheck) and the post-treatment model	Maxillary and mandibular canine ROTATION	Average time:7.2 months Aligner change every 2–3 weeks	Comparison between the amount of rotation predicted and the amount of rotation actually achieved
Simon et al. [21]	Retrospective CCT	Private practice-Cologne (Germany)	30 patients (11 M–19 F) reduced to 26. Age: 32.9 3 configurations: -Incisors with attachments or power-ridges -Premolars with or without attachments -Molars with or without attachments	Incisor TORQUE: -Horizontal ellipsoid attachment or power ridge Premolar DEROTATION: -Optimized Attachment or no attachment Molar DISTALIZATION: -Horizontal Attachment Bevelled in gingival direction or no attachment	-Superimposition between the initial situation (T1) and the final stage of ClinCheck (Clin T2) -Superimposition between the initial situation (T1) and the actual post-treatment condition (T2)	-Incisor TORQUE > 10° -Premolar DEROTATION > 10° -Molar DISTALIZATION > 1.5 mm	Number of aligners: 18 aligners on average	-Comparison between (T2-T1) and (ClinT2-T1) to evaluate treatment efficacy with or without attachments and power ridges - Analysis of the accuracy of premolar derotation according to the staging (degree of derotation per aligner) and to the total amount of predicted movement

**Table 3 materials-15-00533-t003:** Results of risk of bias evaluation performed for clinical studies according to the Downs and Black scale tool.

Study	Reporting	External Validity	Bias	Confounding	Power	Overall	Risk of Bias *
	0–11	0–3	0–7	0–6	0–1	0–28	
Dai et al. [23]	10 of 11	1 of 3	3 of 7	3 of 6	0 of 1	17 of 28	Medium
Durrett [20]	6 of 11	1 of 3	2 of 7	3 of 6	0 of 1	12 of 28	Medium
Garino et al. [16]	9 of 11	1 of 3	4 of 7	2 of 6	1 of 1	17 of 28	Medium
Kravitz et al. [22]	10 of 11	1 of 3	4 of 7	3 of 6	0 of 1	18 of 28	Medium
Simon et al. [21]	9 of 11	1 of 3	3 of 7	3 of 6	0 of 1	16 of 28	Medium

* Risk of Bias—High (0–8); Medium (9–18); Low (19–28).

**Table 4 materials-15-00533-t004:** Analysis of the results (grouped according to the type of movement).

	Study	Type of Study	Aim	Study Design	Results
Anterior B-L tipping/Root torque	Simon et al. [21]	Retrospective CCT	Verification of the effectiveness of the Invisalign treatment by comparing clin-check with the obtained results. Analysis of the influence of attachments and power-ridges, patient compliance and staging (amount of movement per aligner) on treatment efficacy.	30 patients (11 M–19 F) reduced to 26. Age: 32.9 3 configurations: -Incisors with attachments or power-ridges -Premolars with or without attachments -Molars with or without attachments	Incisor torque showed positive results, both with an horizontal ellipsoid attachment on upper central incisors and with power ridges. A torque loss (up to 50%) is a common finding during incisors retraction.
	Garino et al. [16]	RCT	Verification of the influence of the number of attachments on the amount of upper molar distalization	30 patients with class II malocclusion (12 M–18 F) Age: 30.5 3 configurations: -Group C1—5 attachments per quadrant -Group C2—3 attachments per quadrant -Group C—Control	The greatest distalization of central incisors was obtained in C1 group.
	Dai et al. [23]	Case Series	Comparison between predicted and achieved tooth movements of maxillary first molars and central incisors in extraction cases treated with Invisalign.	30 patients (4 M–26 F) Age: 19.4 ± 6.3 First premolar extraction cases treated with Invisalign. Three variables considered: - Age -Type of attachment -Initial crowding	In case of anchorage loss of posterior teeth, the amount of incisors bodily movement in lingual direction was lower than expected, while inclination increased in the same direction. In particular, 3 mm vertical rectangular attachment, located on the maxillary first molar, showed the least efficacy in anchorage control, compared to horizontal attachments (3 or 5 mm) and optimized G6 attachments.
Intrusion	Durrett [20]	RCT	Analysis of the influence of attachments during: -canine and premolar rotation; -incisors, canines and premolars intrusion and extrusion. Comparison between different attachment configurations.	99 patients reduced to 86 Age: 18+ 6 configurations: -5 groups with different attachments -1 control group (without attachments) 2 groups: -Extraction and non-extraction cases	All the attachment groups showed a greater efficiency compared to the control group. The greatest efficiency and degree of correlation was shown by group F, characterized by a vestibular attachment with a poliedric shape.
	Dai et al. [23]	Case Series	Comparison between predicted and achieved tooth movement of maxillary first molars and central incisors in extraction cases treated with Invisalign.	30 patients (4 M–26 F)Age: 19.4 ± 6.3First premolar extraction cases treated with Invisalign. Three variables considered:- Age- Type of attachment- Initial crowding	First molars achieved greater intrusion than predicted. The group with the optimized G6 attachments showed a greater difference between predicted and achieved tooth movement compared to the other groups.
Extrusion	Durrett [20]	RCT	Analysis of the influence of attachments during: -canine and premolar rotation; -incisors, canines and premolars intrusion and extrusion. Comparison between different attachment configurations.	99 patients reduced to 86 Age: 18+ 6 configurations: -5 groups with different attachments -1 control group (without attachments) 2 groups: -Extraction and non-extraction cases	No statistically significant differences were found among the analyzed groups The group with an ovoid attachment on the buccal face, showed the greatest efficiency. However, the small number of the sample didn’t allow to draw clear conclusions.
Rotation	Simon et al. [21]	Retrospective CCT	Verification of the effectiveness of the Invisalign treatment by comparing clin-check with the obtained results. Analysis of the influence of attachments and power-ridges, patient compliance and staging (amount of movement per aligner) on treatment efficacy.	30 patients (11 M–19 F) reduced to 26. Age: 32.9 3 configurations: -Incisors with attachments or power-ridges -Premolars with or without attachments -Molars with or without attachments	No significant differences were found among the analyzed groups. The effectiveness was reduced for predicted rotations greater than 15° and for movements greater than 1.5° per aligner.
	Kravitz et al. [22]	Prospective CCT	Evaluation of the influence of attachments or IPR on canine rotation (comparison between predicted and achieved results)	38 pazients reduced to 31 (13 M–18 F) Age: 29.4 3 configurations: -Attachment only group (AO) -Interproximal-reduction only group (IO) -Group without attachments (N)	No significant differences were found between groups with and without attachments. The IO group performed better than the other ones, because it allowed the creation of space for movement.
	Durrett [20]	RCT	Analysis of the influence of attachments during: -canine and premolar rotation; -incisors, canines and premolars intrusion and extrusion. Comparison between different attachment configurations.	99 patients reduced to 86 Age: 18+ 6 configurations: -5 groups with different attachments -1 control group (without attachments) 2 groups: -Extraction and non-extraction cases	Conflicting results were found between groups that required or not required a “reboot”. In the non-rebooted patients, group C (without attachments) achieved a greater degree of rotation than group F (with a vestibular attachment). In the rebooted patients, on the other hand, the attachment groups were more effective than the control one, except for the group with a vestibular and a lingual attachment. Larger attachments with sharper edges seem to be more effective.
M-D Movement	Simon et al. [21]	Retrospective CCT	Verification of the effectiveness of the Invisalign treatment by comparing clin-check with the obtained results. Analysis of the influence of attachments and power-ridges, patient compliance and staging (amount of movement per aligner) on treatment efficacy.	30 patients (11 M–19 F) reduced to 26. Age: 32.9 3 configurations: -Incisors with attachments or power-ridges -Premolars with or without attachments -Molars with or without attachments	Molar distalization was more effective than the other movements, regardless to the use of attachments (average accuracy of 88.4% for the attachment group and 86.9% for the group without attachments).
	Dai et al. [23]	Case Series	Comparison between predicted and achieved tooth movements of maxillary first molars and central incisors in extraction cases treated with Invisalign.	30 patients (4 M–26 F)Age: 19.4 ± 6.3 First premolar extraction cases treated with Invisalign. Three variables considered:- Age- Type of attachment- Initial crowding	The study highlighted a loss of posterior anchorage: the 3-mm vertical rectangular attachment showed the worst clinical outcome. On the other hand, the G6-optimized attachments and the horizontal rectangular ones seem to be more effective in counteracting mesial tipping.
	Garino et al. [16]	RCT	Verification of the influence of the number of attachments on the amount of upper molar distalization	30 patients with class II malocclusion (12 M–18 F) Age: 30.5 3 configurations: -Group C1—5 attachments per quadrant -Group C2—3 attachments per quadrant -Group C—Control	The number of attachments seems to play play an important role: group C1 (with attachments bonded on the surface of five teeth) showed greater first molar distalization and central-incisor retraction, compared to the other groups. No significant differences were found regarding the efficacy of second molar distalization among the attachment groups.

## Data Availability

All the data are reported in the present manuscript.

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
