# Peer review of "Effects of Composite Attachments on Orthodontic Clear Aligners Therapy: A Systematic Review"

_materials, 2022, doi:10.3390/ma15020533_

Round 1

Reviewer 1 Report

Dear authors, I would like to take the present moment to congratulate the authors for conducting the present review on composite attachments on orthodontic aligners.

Major limitation

My main concern is related with the wide specter of collected studies and methodologies which basically makes difficult to answer to any specific question. Putting together in the same pool in vivo, in vitro and in silico studies in order to achieve an answer for so many possible aspects creates a lot of noise and avoid precise answer based on multiple studies conducted similarly. As an example, the authors have pooled together randomized controlled trials, case reports, finite element studies and laboratorial studies using plastic models. Which means the authors oscillate from a Level 1 evidence concerning systematic reviews with RCT to the last level when conducting reviews on laboratorial results. I think the way the authors are conducting this review has big flaws and I recommend the authors to reformulate their objectives and look for very clear questions and not get everything you can put your hand on.

So my better advice is to reformulate and include only clinical studies. I believe that there are not many clinical studies, which may mean that what we need is not a systematic review but clinical studies.

Here goes a few of my considerations:

KEYWORDS:

Please add “systematic review” to it

INTRODUCTION

The aim sentence should be reformulated. Although it does not follow the Prisma format, at least it should be equivalent. The objective of a systematic review is not necessary to review all the variables under the sun. May the author be clearer regarding what are the question to be answered with the present review?

METHODS

Did the author conducted any manual search? Or only electronic and contact with authors?

The “Inclusion and exclusion” criteria are extremely limited. Basically the authors aimed to find anything that had composite attachments, is that it? What was the study designs accepted? Any specific brands involved? Any specific methods involved? Any specific follow up involved? Any specific outcome involved? What were the authors looking for? Only composite attachments? The authors should be more clear in what are their goals.

Regarding the sentence “Two review authors (C.D. and L.R.) independently performed the preliminary study selection based on titles and abstracts evaluation. The same two reviewers then proceeded to the analysis of the full text, in order to finally select the included articles.The results of the two independent study selections were compared. Cohen's Kappa was calculated to determine the concordance between the two reviewers.”, which results were compared, and how were they scored so they could have been assessed using Cohen’s Kappa?

I have strong reserves regarding the paragraph “Data extraction process”. So, it seam the authors have added together what they define as “clinical studies” and “experimental studies”. I believe it is important to define clearly both groups so one can understand if all should be included in the same pool of studies to be included in a systematic review. Than, which forms were used? Any previously documented or something done for the present study? Which characteristics were assessed and how were they scored so they could have been assessed using Cohen’s Kappa?

RESULTS

The authors mention other sources, which sources are we talking about?

So, the authors have added to the same pool of the included studies: case reports, finite element studies and laboratorial studies using plastic models. I see this as a strong limitation.

Regarding Table 5… did the authors conducted on the included case report the same risk assessment of the RCTs? The study designs are too different to be evaluated the same way.

DISCUSSION

The Discussion is too long, should be somehow more summarized

Level of evidence of the Review is missing

Author Response

Dear Reviewers,

We, as authors, would like to thank you for your comments. We think they helped us improving our manuscript and overcoming its shortcomings. We marked with red color the answers to your remarks.

My main concern is related with the wide specter of collected studies and methodologies which basically makes difficult to answer to any specific question. Putting together in the same pool in vivo, in vitro and in silico studies in order to achieve an answer for so many possible aspects creates a lot of noise and avoid precise answer based on multiple studies conducted similarly. As an example, the authors have pooled together randomized controlled trials, case reports, finite element studies and laboratorial studies using plastic models. Which means the authors oscillate from a Level 1 evidence concerning systematic reviews with RCT to the last level when conducting reviews on laboratorial results. I think the way the authors are conducting this review has big flaws and I recommend the authors to reformulate their objectives and look for very clear questions and not get everything you can put your hand on.

So my better advice is to reformulate and include only clinical studies. I believe that there are not many clinical studies, which may mean that what we need is not a systematic review but clinical studies.

Here goes a few of my considerations:

KEYWORDS:

Please add “systematic review” to it

We added it

INTRODUCTION

The aim sentence should be reformulated. Although it does not follow the Prisma format, at least it should be equivalent. The objective of a systematic review is not necessary to review all the variables under the sun. May the author be clearer regarding what are the question to be answered with the present review?

We replaced the original sentence with: “The aim of this systematic review is to highlight the differences between orthodontic cases treated with and without attachments and to clarify what is the best shape, size, number and position of attachments for each specific orthodontic movement.”

METHODS

Did the author conducted any manual search? Or only electronic and contact with authors?

We did not conduct manual search. This aspect was clarified in the manuscript text.

The “Inclusion and exclusion” criteria are extremely limited. Basically the authors aimed to find anything that had composite attachments, is that it? What was the study designs accepted? Any specific brands involved? Any specific methods involved? Any specific follow up involved? Any specific outcome involved? What were the authors looking for? Only composite attachments? The authors should be more clear in what are their goals.

We aimed to take in consideration any type of evidence observed in the literature regarding the influence of composite attachments on orthodontic treatments, narrowing the field based on the type of study (only clinical trials), the type of attachment material (only composite) and the presence of comparison groups. The heterogeneity of the studies does not allow making a meta-analysis, but we believe it does not preclude the possibility of carrying out a good systematic review.

Regarding the sentence “Two review authors (C.D. and L.R.) independently performed the preliminary study selection based on titles and abstracts evaluation. The same two reviewers then proceeded to the analysis of the full text, in order to finally select the included articles.The results of the two independent study selections were compared. Cohen's Kappa was calculated to determine the concordance between the two reviewers.”, which results were compared, and how were they scored so they could have been assessed using Cohen’s Kappa?

All the retrieved articles were independently evaluated by two review authors (C.D. and L.R.). The following data were calculated for each selection process: number of articles that both the review authors agreed to include, number of articles that both the review authors agreed to exclude, number of articles that only the first review author wanted to include, number of articles that only the second review author wanted to include. This data was used to compute Cohen’s Kappa.

I have strong reserves regarding the paragraph “Data extraction process”. So, it seam the authors have added together what they define as “clinical studies” and “experimental studies”. I believe it is important to define clearly both groups so one can understand if all should be included in the same pool of studies to be included in a systematic review. Than, which forms were used? Any previously documented or something done for the present study? Which characteristics were assessed and how were they scored so they could have been assessed using Cohen’s Kappa?

We excluded experimental studies and removed related tables. Table 2 was specifically created for this study, selecting the most relevant characteristics of the included clinical studies. Every author filled a specific extraction form for each study. In the final stage, the two review authors compared the results of their work and registered any discordant information. The following data were calculated for each study: number of characteristics that both the review authors consistently reported and number of characteristics that authors inconsistently reported. The level of agreement was calculated based on the percentage of consistent findings, compared to the total amount of characteristics comparisons.

RESULTS

The authors mention other sources, which sources are we talking about?

Other sources are represented by OATD (Open Access Theses and Dissertations). This source was mentioned in the manuscript text among the used electronic sources.

So, the authors have added to the same pool of the included studies: case reports, finite element studies and laboratorial studies using plastic models. I see this as a strong limitation.

We excluded experimental studies evaluating only clinical studies.

Regarding Table 5… did the authors conducted on the included case report the same risk assessment of the RCTs? The study designs are too different to be evaluated the same way. 

We used Downs and Black scale which include questions about randomization. It is validated to be used with randomized and non-randomized clinical studies. Downs and Black scale gave to the RCTs a higher score for indicating a lower risk of bias.

DISCUSSION

The Discussion is too long, should be somehow more summarized

Discussion was summarized and reduced

Level of evidence of the Review is missing

The level of evidence was reported in the manuscript text at the end of the discussion section.

Reviewer 2 Report

This manuscript reviews the effects of composite attachments used in conjunction with orthodontic aligners. Despite its length, the article is well-structured and was a pleasure to read. The concise writing and the logical flow of the text make the manuscript very reader-friendly. The Introduction is comprehensive and understandable to a wider circle of readers, even outside the field of orthodontics, which is otherwise very specialized. The methodology follows the PRISMA statement and has been previously published. The search strategy is comprehensible and described in detail. By using strict selection criteria, the authors included 24 studies, which is a reasonable number for quantitative analysis. The Results are very extensively presented (maybe some of the tables could be supplementary material, but this is not necessary), giving the reader an insight into all relevant data extracted from the collected studies. The Discussion section is also quite extensive, however, this is necessary given the abundance of data that was extracted. The narrative of the Discussion section is clear to the non-specialist and also emphasizes clinical relevance of the reviewed findings. The Conclusion section is written as a bullet-point summary of main take-home messages, which makes a nice ending for the lengthy article.  Overall, this is an article with a significant contribution and I would like to congratulate the authors for their efforts.

Author Response

We thank the reviewer for his/her positive comments

Reviewer 3 Report

Dear authors

This review manuscript was aimed to summarize the effects of composite attachments on orthodontic clear aligners therapy. This topic in this review is clinically relevant and interesting for clinician related wiht orthodontic treatment. Unfortunately, this manuscript is not appropriately organized as for formal systematic review under newest guidelines (PRISMA 2020). I ask the authors to conduct review work, to describe method, and to present results according to the newest international standard guideline; PRISMA 2020 (http://www.prisma-statement.org). Please note that online pre-registration on PROSPERO (https://www.crd.york.ac.uk/prospero/) is required for formal systematic review to be published in Materials. The re-submitted manuscript should include a completed PRISMA 2020 checklist. 

Author Response

This review manuscript was aimed to summarize the effects of composite attachments on orthodontic clear aligners therapy. This topic in this review is clinically relevant and interesting for clinician related wiht orthodontic treatment. Unfortunately, this manuscript is not appropriately organized as for formal systematic review under newest guidelines (PRISMA 2020). I ask the authors to conduct review work, to describe method, and to present results according to the newest international standard guideline; PRISMA 2020 (http://www.prisma-statement.org).

The Manuscript was reviewed according to the most recent PRISMA recommendation (PRISMA 2020).

 Please note that online pre-registration on PROSPERO (https://www.crd.york.ac.uk/prospero/) is required for formal systematic review to be published in Materials.

PROSPERO pre-registration was originally performed. (https://www.crd.york.ac.uk/prospero/display_record.php?ID=CRD42020150671 with the registration number CRD42020150671. )

The re-submitted manuscript should include a completed PRISMA 2020 checklist.

PRISMA 2020 checklist was edited and submitted with the previsioned manuscript.

Round 2

Reviewer 1 Report

Dear authors, I have no more concerns. Thank you

Author Response

Dear Reviewer

thank you so much

Reviewer 3 Report

Ref. 18 showed 2009. This is not newest. 

Author Response

Reference 18 has been changed